# cGPS Record of Active Extension in Moroccan Meseta and Shortening in Atlasic Chains under the Eurasia-Nubia Convergence

**DOI:** 10.3390/s23104846

**Published:** 2023-05-17

**Authors:** Ahmed Chalouan, Antonio J. Gil, Ahmed Chabli, Kaoutar Bargach, Hoda Liemlahi, Khalil El Kadiri, Víctor Tendero-Salmerón, Jesús Galindo-Zaldívar

**Affiliations:** 1Faculty of Sciences, Mohammed V University in Rabat, Rabat 10000, Morocco; chalouan@yahoo.com; 2Departamento de Ingeniería Cartográfica, Geodesia y Fotogrametría, Universidad de Jaén, 23071 Jaén, Spain; ajgil@ujaen.es; 3Centre Régional des Métiers de l’Éducation et de la Formation de Rabat, Rabat 10000, Morocco; ahmedchabli308@gmail.com; 4Geo-Biodiversity and Natural Heritage Laboratory (GEOBIO), Scientific Institute, Mohammed V University in Rabat, Rabat 10000, Morocco; kbargach50@gmail.com; 5École Normale Supéreure, Université Abdelmalek Essaadi, Martil 93150, Morocco; hodaliemlahi@yahoo.fr; 6Faculté des Sciences, Université Abdelmalek Essaadi, Tetouan 93000, Morocco; khalilelkadiri@gmail.com; 7Departamento de Geodinámica, Universidad de Granada, 18071 Granada, Spain; vtendero@ugr.es; 8Instituto Andaluz de Ciencias de la Tierra, CSIC-Universidad de Granada, 18100 Armilla, Spain

**Keywords:** cGPS measurements, Nubian plate boundary, westernmost Mediterranean, slow active extensional tectonics, anomalous mantle

## Abstract

The northwest-southeast convergence of the Eurasian and Nubian (African) plates in the western Mediterranean region propagates inside the Nubian plate and affects the Moroccan Meseta and the neighboring Atlasic belt. Five continuous Global Positioning System (cGPS) stations were installed in this area in 2009 and provide significant new data, despite a certain degree of errors (between 0.5 and 1.2 mm year^−1^, 95% confidence) due to slow rates. The cGPS network reveals 1 mm year^−1^ North/South shortening accommodated within the High Atlas Mountains, and unexpected 2 mm year^−1^ north-northwest/south-southeast extensional-to-transtensional tectonics within the Meseta and the Middle Atlas, which have been quantified for the first time. Moreover, the Alpine Rif Cordillera drifts towards the south-southeast against its Prerifian foreland basins and the Meseta. In this context, the geological extension foreseen in the Moroccan Meseta and Middle Atlas agrees with a crustal thinning due to the combined effect of the anomalous mantle beneath both the Meseta and Middle-High Atlasic system, from which Quaternary basalts were sourced, and the roll-back tectonics in the Rif Cordillera. Overall, the new cGPS data provide reliable support for understanding the geodynamic mechanism that built the prominent Atlasic Cordillera, and reveal the heterogeneous present-day behavior of the Eurasia-Nubia collisional boundary.

## 1. Introduction

The analysis of tectonic displacements is essential for the reconstruction of the Earth’s geodynamics, and is particularly relevant for understanding active tectonic processes linked to internal geological hazards. Faulting, volcanism, and seismicity mainly occur along tectonic plate boundaries, which are the most sensitive regions that have undergone catastrophic geological events. GPS (Global Positioning System) observations in permanent monuments constitute the most accurate technique for revealing active tectonic deformations in mountain ranges when compared to classical satellite and surface geodetic observations [1]. Continuous GPS (cGPS) records increase the accuracy achieved by non-permanent GPS networks and allow for obtaining GPS positions with velocity differences of submillimetric deformations between stations that are tens of kilometers apart. However, the described technique is more expensive than others because it requires instrumentation and permanent technical support, which greatly reduce the number of instruments that can be installed. A preliminary geological survey is needed to carefully select the key structural locations that enable cGPS stations to consistently reveal the relative motion of the interacting tectonic domains. The selected location should also provide good conditions for the safe preservation of equipment, surveillance to avoid vandalism, and an open horizon for optimal satellite coverage observations.

The Rif and Atlas cordilleras are the most prominent mountain ranges at the northwestern-most margin of the Nubian (African) plate (Figure 1). These cordilleras are separated by the Moroccan Meseta, an uplifted region formed by Variscan basement rocks [2] and hosting Cenozoic-to-Quaternary intramontane, satellite or foredeep basins (Gharb, Saïss, Guercif, Missour, Moulouya, Tadla, Bahira, Ahouz, Souss and Ouarzazate).

Plate tectonic reconstructions since the latest Miocene suggest an average convergence rate of 5 mm year^−1^ northwest-southeast (NW-SE) to North-South (N-S) between the Nubian and Eurasian plates in the westernmost Mediterranean [4,5]. Present-day Nubian-Eurasian convergence is also well-constrained by GPS measurements, particularly in the Rif and Betics [6,7,8,9,10]. Although these measurements reveal detailed regional displacement of the Rif Cordillera with respect to Nubia, they do not provide enough resolution in the Meseta and Atlas due to the low deformation rates and because the GPS sites were based on non-permanent observations. To refine the geodetic data and properly understand the present-day behavior of this region, three new permanent stations were built in Morocco as part of the Topo-Iberia project (2009–2013, TAZA, BENI, ERRA) and have been integrated with the available stations of RABT and CEU1 [11].

The aim of this contribution is to present the initial results of continuous GPS (cGPS) data from the 5 key stations located in the weakly deformed northwestward border of the Nubian plate. Indeed, these data reveal unexpected simultaneous extensional and shortening deformations in this convergent plate boundary. Moreover, they provide new insights into how the prominent Alpine Atlasic cordillera was built, the old Variscan Mesetian basement was uplifted, and Quaternary basaltic volcanism concurrently came into being.

## 2. Geological Setting

### 2.1. Tectonic Structure

The Rif cordillera, along with the Betic cordillera, forms the Alpine Gibraltar Arc (Figure 1a), a NW-SE convergent interplate region that borders Eurasia and the Nubian (African) plates in the westernmost Mediterranean. The plate boundary is well-defined to the east, where it continues into the Tell and the Kabylies, and to the west, where it extends into the Atlantic Ocean towards the Azores-Gibraltar fault zone. In contrast, in the Gibraltar Arc, there is a wide deformation zone, irregularly distributed, that reaches more than 300 km wide, and the location of the plate boundary is still under discussion [10].

The Gibraltar Arc has been emplaced towards the west between the main plates since the Cenozoic period [12]. The Alboran Sea constitutes the westernmost part of the Mediterranean Sea and is floored by continental crust. It represents the main Neogene basin surrounded by the Betics and Rif cordilleras, which are the northern and southern branches of the tectonic arc, respectively. The westward emplacement of the Rif has favored the development of the large transcurrent sinistral Jebha (JF) and Nekor (NF) faults (Figure 1). This westward emplacement of the tectonic arc has developed two curved Neogene-to-Quaternary foreland basins in the Rif, the Gharb towards the west and the Saïss towards the south. Their mountain fronts show evidence of recent activity affecting the Quaternary sedimentary rocks, as seen in Fès (F, Figure 1) [13]. There, local non-permanent GPS networks have quantified the active deformation rates [9].

The Moroccan Meseta has been considered as the foreland of the Alpine Rif Cordillera, and is therefore relatively attached to the Nubian Plate. It is a piece of the Variscan Belt, mainly formed by metamorphic and igneous rocks that were deformed during the late Paleozoic and locally covered by undeformed Mesozoic-Cenozoic sedimentary cover [14]. It is named the Meseta because it has relatively uplifted topography with generally low reliefs. It is formed mainly by the Western and Central Meseta and separated by the Middle Atlas from the Eastern Meseta.

In contrast to the Rif, the Atlas system has been considered as a series of intraplate ranges [3,15,16] although these ranges are characterized as the highest relief mountains at the plate boundary. The Atlas system is formed by inverted elongated Mesozoic basins filled by sedimentary series similar to those found around the Mediterranean Alpine cordilleras. These deposits were located on the passive continental margins and areas of continental crustal thinning around the former Tethys Ocean. The Atlas basins were surrounded by Variscan low deformed blocks (Eastern and Western Moroccan Meseta) and the Precambrian-to-Paleozoic Anti-Atlas. The pre-Mesozoic blocks were also covered by Mesozoic to Cenozoic undeformed deposits that now constitute the “plateaus”. Since the Neogene, compressive deformation due to the Eurasian-African convergence has affected these elongated, crustal thinned weak zones where the basins were located. Finally, basin inversion has occurred and the elongated cordilleras have developed along the former sedimentary basins. The Atlas is composed of two main branches, the High Atlas of east-northeast/west-southwest (ENE-WSW) orientation and the Middle Atlas of northeast/southwest (NE-SW) orientation.

The High Atlas (Figure 1b) is located along the transcurrent and reverse faults running between the Variscan Maghrebian realm (Moroccan Meseta) and the northern margin of the West African shield, which is represented by the Anti-Atlas, where Precambrian and Paleozoic rocks outcrop. The High Atlas is divided into western, central, and eastern sections. The western High Atlas reaches the maximum elevation and is made up of outcropping deformed pre-Mesozoic rocks, including metamorphic and igneous rocks. It is bounded by compressive tectonic structures that separate the mountains and the surrounding basements, and favors the development of the Haouz Basin towards the north and the Souss Basin towards the south in their foot blocks. In contrast, the Central and Eastern High Atlas only expose the Mesozoic carbonate sedimentary series. To the south, reverse faulting forms the boundary of the Ouarzazate Basin, while to the north, the Tadla, Moulouya, and Missour basins are found.

The Middle Atlas is a NE-SW alpine chain that separates the Variscan-derived eastern Meseta from the western one. This chain is itself divided by the 200 km long North Mid-Atlas Fault (NMAF), oriented NE-SW, into the Tabular Middle Atlas (also known as the “Causse,” to the west) and the folded Middle Atlas (to the east). The latter hosts thick early-to-mid-Jurassic marl-limestone sequences that are folded into several NE-SW oriented anticlines and synclines separated by decollement faults in the same direction. The folded Middle Atlas connects the Central High Atlas and the eastern Rif (see Figure 1b). It was formed by the Alpine reactivation of the Variscan thrust front that pushed the eastern Meseta over the western one (see Figure 1). To the south and east, this deformation zone marks the boundary of the Neogene Mouloya Basin, which is sandwiched between the High and Middle Atlas, as well as the Missour and Guercif basins that separate the Middle Atlas from the Eastern Moroccan Meseta.

The geodynamic evolution of the region implies the occurrence of several episodes of volcanism, with one of the most notable being the eruption of Quaternary basaltic rocks mainly located in the Middle Atlas [17]. These eruptions indicate that there is melting at the base of the crust of this intraplate mountain belt. Volcanic rocks extend to the Central Meseta and even the Saïss Basin (Figure 1b).

### 2.2. Seismological and Active Tectonic Setting

Deformations at plate boundaries are accommodated by active tectonic structures, which include folds and faults. Active faults may move continuously by creep or suddenly, producing earthquakes that are determined by the rheological behavior of fractured rocks [18]. Seismicity constitutes the main evidence of the location of the main active tectonic structures. Moreover, the earthquake focal mechanism allows for the revelation of faulting features and the stress regime of each region.

The distribution of seismicity in Morocco (Figure 2) is heterogeneous. Most of the seismicity occurs in a main cluster in the eastern Rif Cordillera, where earthquake series are mainly related to strike-slip faulting in the Al Hoceima area (Figure 2b). This is a very active deformation area of the plate boundary where basement blind faults develop [19]. Moreover, seismicity is also very active towards the western and southern boundaries of the Rif Cordillera. Quaternary-to-recent deformations are clearly recorded in the southern Rif front [9]. While the geological structures of these regions support the presence of shallow compressive structures with westward and southward vergences [9,20], the earthquake focal mechanisms suggest the main activity of deep strike-slip, with dominant N-S to NW-SE compression and orthogonal extension (see focal mechanisms between Tanger and Meknes, Figure 2b).

Simultaneously, the southern margin of the Saïss basin, attached to the Middle Atlas and to the northern margin of the western Meseta, shows geological brittle structures that support a recent extensional regime [9,13,21,22,23,24,25]. The Moroccan Meseta is affected by moderate, heterogeneously distributed seismicity which is most intense towards the Central Meseta (Figure 2a). The Folded Middle Atlas is bounded by the North and South Middle Atlasic faults, with maximum seismicity concentration close to the North Middle Atlasic Fault (NMAF, Figure 2) [2]. The earthquake focal mechanisms close to this main fault show variable features (Northeast of Beni Mellal, Figure 2b), including strike-slip and reverse faulting with N-S-to-north-northwest/south-southeast (NNW-SSE) compression, but also local strike-slip with NW-SE-to-E-W extension. The NMAF has been interpreted by field geological observations as a reverse fault with sinistral strike-slip [22]. This area is also affected by Quaternary basaltic volcanism. In contrast, the South Middle Atlasic Fault (SMAF, Figure 2) has lower seismicity than the NMAF and is characterized by a reverse earthquake focal mechanism, clearly evidencing NW-SE compression and the reverse character of the fault.

Active seismicity also occurs, but at a lower intensity, in the High Atlas, bounded by the South and North High Atlasic faults. It becomes very scarce in the Anti-Atlas (Figure 2a), which is presumed to be part of the stable Nubian plate. The earthquake focal mechanisms of the South High Atlasic Fault (SAF) (South of Beni Mellal, Figure 2b) support the hypothesis that this fault underwent reverse faulting related to the NNW-SSE compression and probably strike slip deformation.

### 2.3. Geodynamic Models

The complex geodynamic setting of the westernmost Mediterranean has been the subject of various tectonic models that attempt to explain the main features of an Alpine tectonic arc at a large convergent plate boundary. In addition, the presence of the highest relief far south of the Alpine ranges, where the plate boundary is expected, is another issue that needs to be addressed by the proposed models.

Two distinct tectonic mechanisms behind the plate convergence have been proposed: (i) in the Rif, an east-dipping lithospheric plate is subducted beneath the Gibraltar Arc [26,27], resulting in a roll-back setting [28], and (ii) in the Atlas Mountains, the lithosphere is abnormally thinned and hot [3,29,30,31,32] due to an ascending asthenospheric dome [33,34], resulting in the thermal uplift of the entire Atlas chain and the emplacement of Quaternary alkaline volcanism [35]. In this context, the presence of anomalous mantle [3] explains the moderate tectonic shortening of the Atlas Mountains despite their unusually high topography, with the highest peak reaching 4167 m a.s.l., while the Rif only reaches 2456 m a.s.l. [3,34] in spite of its comparatively strongest tectonic shortening.

## 3. cGPS Network, Equipment and Data Processing

This study presents the GPS velocity field derived from continuous observations (cGPS) carried out under the Topo-Iberia framework [11]. Three Topo-Iberia sites (TAZA, BENI, ERRA) and two EUREF sites (CEU1, RABT) in significant locations in northwestern Nubia were selected to reveal active tectonics. They are listed below, from south to north (Figure 1, Figure 2 and Figure 3):Errachidia (ERRA) is located over the Anti-Atlas basement and represents a reference for the stable Nubian plate. It is located just close to the active South Atlasic thrust front, where the central High Atlas overrides the Anti Atlas. This station is located on the Mesozoic-Cenozoic plateaus developed on the Precambrian-to-Paleozoic basement. No recent geological deformation occurs southward of this station.Béni Mellal (BENI) is located in the Tadla basin, a foredeep Atlasic structure floored by the southernmost boundary of the western Meseta basement. The station is located close to the active Northern Atlasic Fault thrust front and also close to the North Middle Atlas Fault.Rabat (RABT) is located in the northernmost outcrops of the Western Moroccan Meseta, in a region of scarce seismicity close to the southern margin of the Gharb basin, an Alpine Rifian foreland basin.Taza (TAZA) is a key station located on the easternmost outcrops of the central Meseta, close to the North Middle Atlas Fault and close to the junction with the collisional front of the central southern Rif along the boundary with the Middle-Atlasic basement.Ceuta (CEU1) is located at the northernmost end of the Alpine Rif chain, i.e., in the central part of the Nubia-Eurasia interplate area, and contributes to determining the present-day displacements of the Gibraltar Arc.

The Topo-Iberia cGPS network installation was completed in December 2008, and all the stations have been fully operational since then. The data analysis was performed at three different analysis centers: Real Instituto y Observatorio de la Armada (ROA), the University of Barcelona (UB), and the University of Jaen (UJA). Several approaches to processing GPS data were carried out using different software [11]. In this paper, the cGPS data covering the 2004–2012 timespan have been used for sites CEU1 and RABT, and the data covering 2008–2012 for BENI, ERRA, and TAZA. After this period, the record of Topo-Iberia stations became discontinuous due to the end of the project and the irregular economic and technical support.

The cGPS data processing followed the standard method used by the University of Jaen [36]. Initially, the CGPS data underwent a quality analysis. Subsequently, Bernesse software [37], with options shown in [11], was employed to carry out the data processing, which resulted in a daily GPS network solution in a loosely constrained reference frame. Next, the daily network solutions were transformed into ITRF2005 by minimal constraints, estimating translations and scale parameters. Then, the estimation of the crustal velocity field was computed from the ITRF2005 time series using the software NEVE, which managed the complete stochastic model [38,39]. The GPS-derived site velocities and uncertainties in the ITRF2005 reference frame are shown in Table 1. A more effective representation of the velocity field estimated is thought to determine the residual velocities with respect to the stable Eurasian plate and take into account the Euler pole of the Eurasian plate [40] (Figure 3).

**Figure 3 sensors-23-04846-f003:**
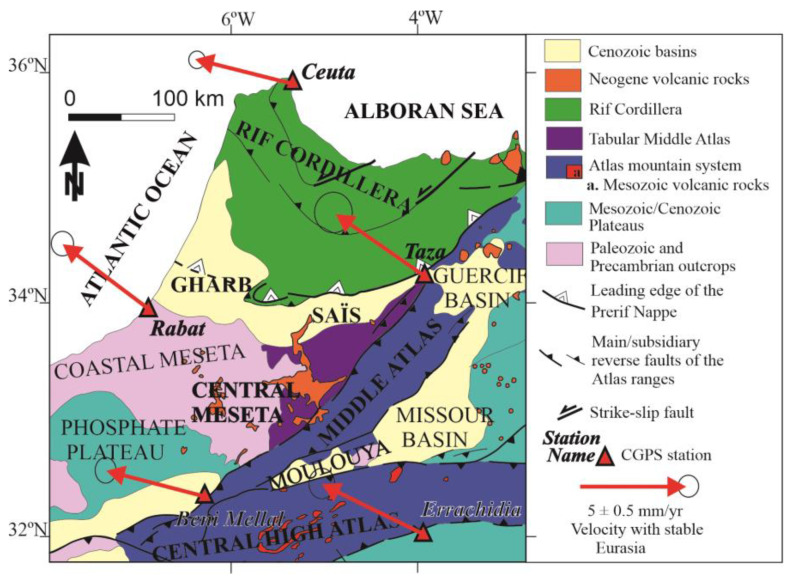
Close-up view of the study area geological map that includes the residual cGPS velocities with respect to stable Eurasia with error ellipses of 95% confidence. The corresponding values for this figure and the following figures have been obtained through computerization using the Bernese software [11] and taking into account the Euler pole of the Eurasian plate [40].

## 4. Velocity Rates from cGPS Stations

The absolute velocities obtained and the related errors are presented in Table 1, with displacement towards the NE. However, their relative displacements are the significant values that allow us to determine the tectonic deformation rates of the structures in the region.

The residual velocities with respect to fixed Eurasia of each GPS station are presented in Table 1 and Figure 3. All the stations have a displacement towards the WNW with respect to stable Eurasia (BENI, 4.8; CEU1, 4.6; ERRA, 5.2; RABT, 5.1; TAZA, 5.1 mm year^−1^) with rates ranging between 4.6 mm year^−1^ for CEU1 and 5.2 mm year^−1^ for ERRA, and they characterize the regional NW-SE plate convergence in the area.

Moreover, in order to establish the relative motion with respect to the most stable Nubian plate, we have determined the relative motion of all these stations with respect to the ERRA site (Figure 4). BENI has a southward displacement of 1.1 mm year^−1^, orthogonal to the elongated Central High Atlas Mountains. RABT and TAZA, located in the northern and northeastern Meseta, respectively, have roughly similar displacement patterns towards the northeast of 1 mm year^−1^ and 0.8 mm year^−1^. These data highlight a significant northeast-southwest extension in the central Meseta of close to 2 mm year^−1^. SSW displacement of CEU1 is 1.1 mm year^−1^, and this determines a shortening with respect to RABT and TAZA, but this displacement is roughly similar in trend and magnitude to that of the BENI site.

Finally, in order to better present the deformation of the Meseta, the BENI station has been considered as the reference (Figure 5) and both RABT and TAZA show notable displacement towards the NNE, while CEU1 is affected by a very low relative displacement towards the SSE.

## 5. Discussion

### 5.1. Analysis of the cGPS Tectonic Displacements

The new data obtained from key sites by cGPS open new perspectives for our understanding of the simultaneous development of both the alpine Rif and the intraplate Atlasic cordilleras. The Nubian-plate northern boundary in the westernmost Mediterranean segment underwent continental collision with distributed active deformation and seismicity [10]. Although error ellipses are not small, a displacement pattern can be attempted for the first time in this region.

The general pattern of the selected sites when Eurasia is fixed (Table 1 and Figure 3) agrees with the expected northwestward displacement of Nubia in respect to Eurasia at rates close to 5 mm year^−1^ [4,5,6,7,8,9]. However, these results reveal a heterogeneous behavior.

In order to unveil this issue, the displacement vectors are considered in respect to stable Nubia (Errachidia, Figure 4). The southward displacement of Beni Mellal, at about 1 mm yr^−1^ rate, which is orthogonal to the High Atlas and its bounding faults (the South High Atlas and North High Atlas fault), gives evidence that a significant part of the deformation related to the plate boundary is occurring in the High Atlas. This convergence trend is also in agreement with the reverse earthquake focal mechanism that occurs in the South High Atlas Fault (Figure 2b). This shortening, together with the underlying anomalous mantle [3,33,34], drives this cordillera to uplift highly in respect to its relative low shortening. In [34], the authors showed, through deep seismic survey and recent Atlasic volcanism, that the Atlasic anomalous topography was conducted during Quaternary times by an ascending hot asthenospheric dome, topped at 25 km depth beneath the High-Moulouya plateau. Microtectonic structures affecting continental deposits surrounding the Atlasic mountains were reactivated during the Quaternary [41,42,43,44] under an NNE/SSW- to-N/S-directed compressional stress regime. These results agree with the convergence obtained by cGPS that now allows for quantifying its related deformation rate.

The comparison of the displacements of Beni Mellal, Rabat, and Taza allows us to constrain the deformation of the presumed stable central Moroccan Meseta. The similar behavior of the deformation rates of Rabat and Taza (Table 1, Figure 3, Figure 4 and Figure 5) invites different interpretations. The easy one is that these locations might belong to the same tectonically stable block located in the northern part of the central Meseta. Nonetheless, their close tectonic setting indicates that Rabat undergoes the extensional effect of the transitional margin between the Meseta and the Gharb basin, whereas Taza is undergoing two combined effects: the southward displacement of the frontal part of the Rif and the extension induced by the eastern end of the Gharb basin (Figure 1, Figure 2, Figure 4 and Figure 5). The displacements resulting from both processes may coincidentally fall in the same rate value. In any case, the relative northeastward displacement of these stations with respect to stable Nubia (Figure 4) and with respect to the southern central Meseta (Figure 5) raises a new line of thinking to understand the present-day behavior of the Meseta.

First, the relative displacement of the northern Meseta (Rabat and Taza) with respect to Beni Mellal provides evidence for very active extensional tectonics in this area of about 2 mm year^−1^ in the NNE-SSW direction (Figure 5 and Figure 6). These results agree with the presence of subtle extensional tectonic deformations envisaged by field geological studies in this region including normal faults [45,46,47,48] and flexures [42]. Moreover, the existence of basaltic volcanism, in addition to the geophysical evidence of the underlying anomalous mantle [3], agrees with the present-day active extension simultaneous to the relief uplift.

The northeastward relative displacement of Rabat and Taza, parallel to the North Middle Atlas Fault (Figure 4 and Figure 5), also raises questions about the present-day kinematics of this major structure. The fault has traditionally been considered to result from sinistral transpressional thrust during Quaternary times based on geological field evidence [9,22,23,46,47,49]. However, the available focal mechanisms along this major structure (Figure 2b) are variable and do not provide a clear answer to this issue. More detailed geodetic research on this area is needed in the future to resolve this apparent inconsistency between field geological and geodetic results.

Further north, in the Rif Cordillera, the relative convergence of Ceuta in respect to Taza and Rabat (Table 1 and Figure 4 and Figure 5) confirms the shortening related to the build-up of the Alpine Rif Cordillera. Previous GPS research [6,7,8,9,10] has already shown evidence of this. The N-S to NNW-SSE trend of convergence with respect to Nubia is in agreement with the similar trend of compression evidenced by the earthquake focal mechanisms of Figure 2b. These results raise questions about the present-day low active westward displacement of the frontal part of the Gibraltar Arc in respect to Nubia, which is substituted by a more intense NNW-SSE convergence in the Rif Cordillera. The southward emplacement of the Rif Cordillera with respect to Nubia determines the highest activity of the southern front of the Cordillera along the northern Saïss basin border, in agreement with geological observations [9,20], rather than in the western border.

### 5.2. Strain Partitioning in the Northern Nubian Plate Boundary: Towards an Integrated Tectonic Model

The significant results evidenced for the first time by the cGPS stations (Figure 2 and Figure 3) constitute a useful reference to constrain the geodynamical models of this region. While a progressive accommodation of the Eurasian-Nubian at a 5 mm yr^−1^ rate, NW-SE convergence was expected in this region, the presence of an unexpected fast extension of 2 mm year^−1^ in the Meseta needs to be considered. Moreover, the roughly southwards displacement of Ceuta (Rif Cordillera) was proved to be roughly similar to the displacement of Beni Mellal (southern Meseta), and the extension of the Meseta was also roughly similar to the convergence of the Rif (between Ceuta and Taza/Rabat). In this setting, a geodynamic model may be proposed for the region based on two coevally intervening processes (Figure 6): (i) the presence of an anomalous mantle below both the High Atlas and Meseta [33,34], and (ii) the roll-back tectonics in the Betic-Rif Cordillera [28] that determine, in the Rif, a relative northeastward displacement of the Variscan basement in respect to the Alpine cover. The combination of these two processes simultaneously results in: (i) extensional crustal bands inside the Gharb basin, the northwestern Meseta, and the Middle Atlas, (ii) the thermal uplift of the High-Atlas, (iii) the basaltic Quaternary volcanism in the Middle Atlas and the neighboring Meseta, and (iv) the shortening of both the High Atlas and the Rif. In this regional setting, the anomalous mantle beneath the Middle and High Atlas remains the driving mechanism behind the development of its higher reliefs compared to those lower in the more shortened Rif.

These data show the heterogeneous behavior of the studied interplate area and open the discussion on whether its active, true, southernmost boundary might be located in the contact between the High Atlas and the Anti-Atlas, instead of being along the classically admitted Rifian thrust front.

## 6. Conclusions

The new continuous GPS (cGPS) data provide reliable support for the first time to understand the geodynamic mechanism that formed the prominent Atlas Cordillera and reveals the heterogeneous present-day behavior of the Eurasia-Nubia collisional boundary.

The results from the cGPS stations located in key tectonic sites inside the northwestern boundary zone of the Nubian plate provide new data on the present-day extensional and compressive deformation of this region. They contribute to revealing the origin of the uplift of the Meseta and Atlas Mountains. The new cGPS data confirm the NW SE Eurasian-Nubian convergence at an overall rate of 5 mm year^−1^, as well as the already well-established convergence in the Alpine Rif Cordillera. However, the present-day southward displacement of the northern Rif in respect to Nubia suggests that active tectonic compressive structures are mainly developed in the southern Rif front while the westward front that delineates the arched shape of the Cordillera becomes of scarce activity. Moreover, these data clearly reveal, for the first time, an NNE-SSW extension of close to 2 mm year^−1^ in the Meseta and possibly in the Middle Atlas. Simultaneously, the relative uplift and the presence of Quaternary basaltic volcanism support the presence of an underlying anomalous mantle, also evidenced by geophysical data. Extensional tectonics may also be favored by the roll-back tectonics occurring in the Rif Cordillera (Figure 6). Southward, convergence of the High Atlas is established for the first time at a rate of 1 mm year^−1^ with an N-S trend, a moderate shortening rate that, together with the underlying anomalous mantle, is responsible for the highest reliefs of the above-mentioned southern plate boundary.

These cGPS data reveal the heterogeneous behavior of the Nubia interplate area and highlight the need to consider the Atlas Mountains as the most prominent Alpine cordillera in this region, where they underline, at the same time, the sharp boundary with stable Nubia.

## Figures and Tables

**Figure 1 sensors-23-04846-f001:**
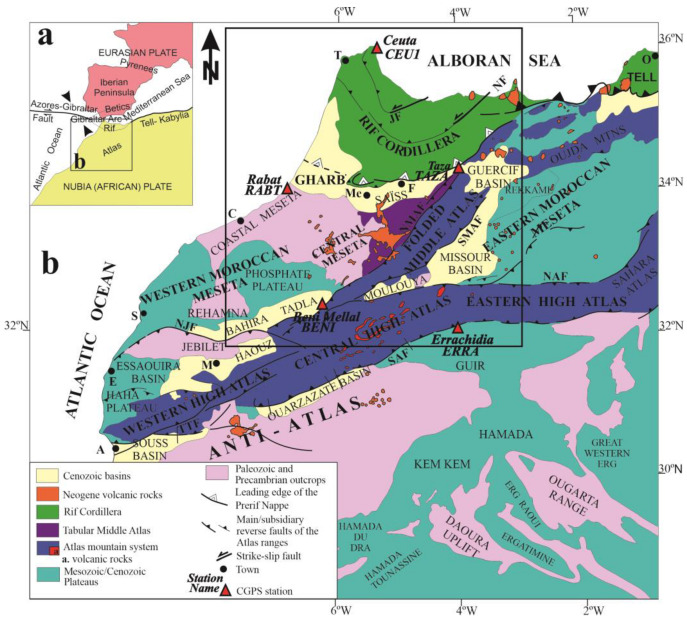
Geological location of the study area. (**a**) Simplified tectonic location of the study area. (**b**) Structural map of northern Morocco (modified from [3]) which shows the main geological units and domains. In this figure and the following ones, names in bold correspond to main topographic and/or geologic features, while those not in bold correspond to smaller or secondary toponymies and geologic features. Abbreviations of faults: JF: Jebha Fault; NAF: North Atlasic Fault; NF: Nekor Fault; NJF: North Jebilet Fault; NMAF: North Mid-Atlasic Fault; SAF: South Atlasic Fault; SNMAF: South Mid-Atlasic Fault. Abbreviations of towns: A: Agadir; C: Casablanca; E: Essaouira; F: Fes; M: Marrakech; Me: Meknes; O. Oran; S: Safi; T: Tanger.

**Figure 2 sensors-23-04846-f002:**
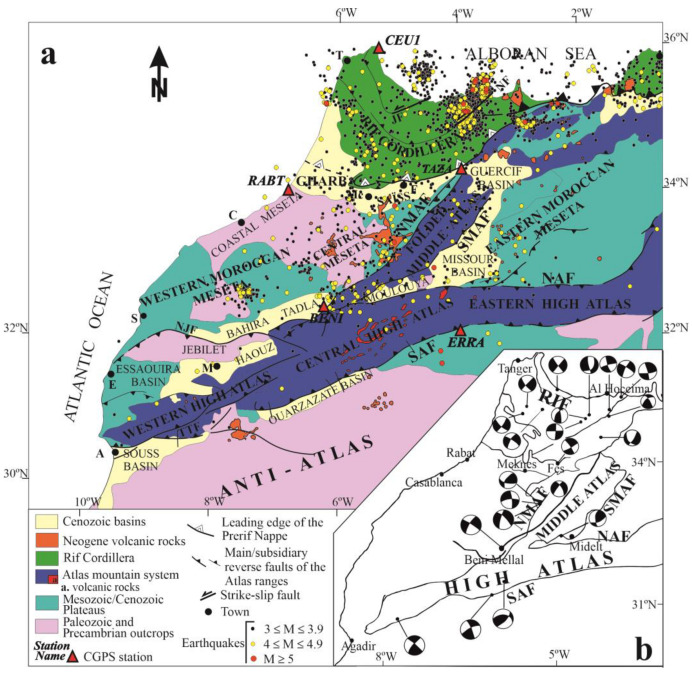
Seismicity and main faults of the study area. (**a**) Seismicity from 1991 to 2022, obtained from IGN (www.ign.es (accessed on 1 February 2023)). Abbreviations of faults: JF: Jebha Fault; NAF: North Atlasic Fault; NF: Nekor Fault; NJF: North Jebilet Fault; NMAF: North Mid-Atlasic Fault; SAF: South Atlasic Fault; SNMAF: South Mid-Atlasic Fault. Abbreviations of towns: A: Agadir; C: Casablanca; E: Essaouira; F: Fes; M: Marrakech; Me: Meknes; O. Oran; S: Safi; T: Tanger. (**b**) Main focal mechanisms for the same period, obtained from IGN (www.ign.es (accessed on 1 February 2023)), which provide information about the types of fault that caused the seismicity and the stress that affects the area.

**Figure 4 sensors-23-04846-f004:**
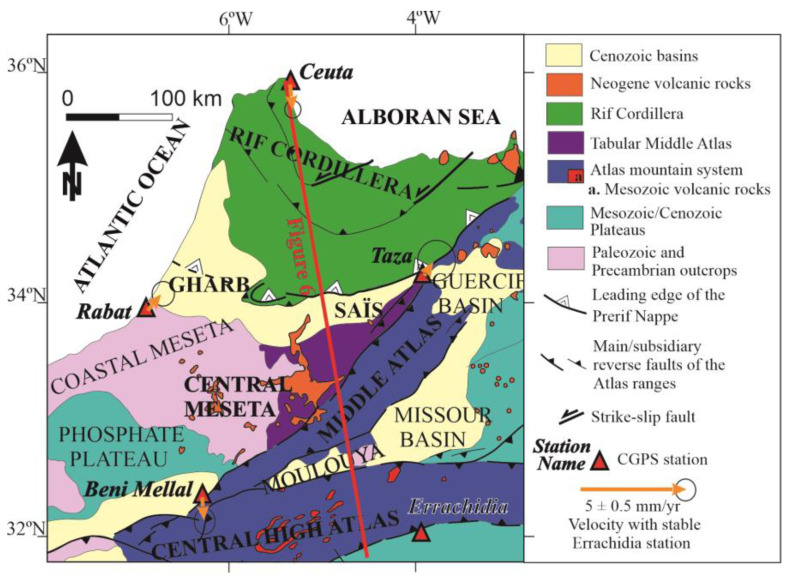
Geological map including residual cGPS velocities respect to the Errachidia station, which can be considered as a representation of Nubia, with error ellipses of 95% confidence. The trace of Figure 6 is represented as a red line on the map.

**Figure 5 sensors-23-04846-f005:**
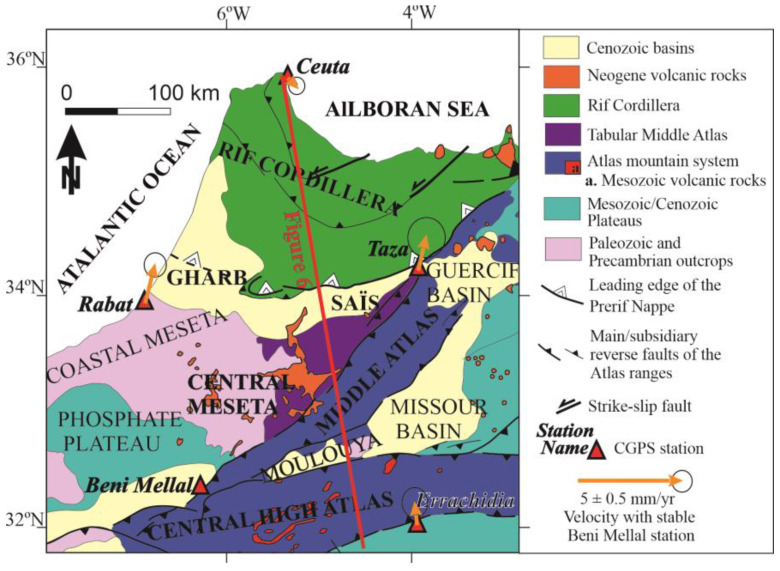
Geological map including residual cGPS velocities respect to the Beni Mellal station, which represents the southernmost sector of Central Moroccan Meseta, with error ellipses of 95% confidence. The trace of Figure 6 is represented as a red line on the map.

**Figure 6 sensors-23-04846-f006:**
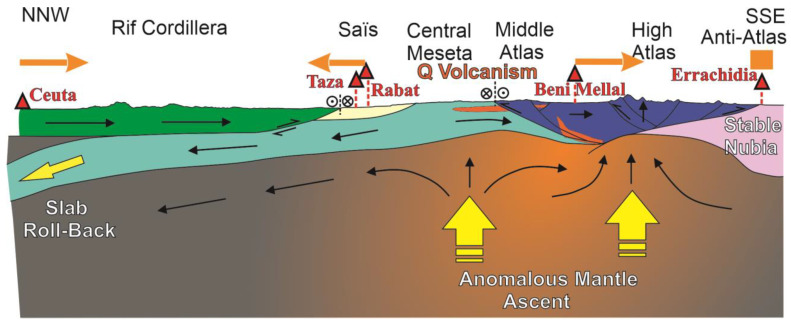
Tectonic sketch along a N-S profile along the transect between the Rif Cordillera and the Anti-Atlas. It illustrates the geodynamic hypothesis for the extension of the Western and Central Moroccan Meseta. The projections of the cGPS stations are depicted. Cross and point, strike-slip faulting.

**Table 1 sensors-23-04846-t001:** Absolute velocities in East and North components from cGPS position time series in ITRF2005 frame and 1σ uncertainties. Residual velocities with respect to Eurasia-fixed reference frame and Errachidia site (Nubia).

Site ID	Latitude (Deg.)	Longitude (Deg.)	Height (m)	Velocity (mm Year^−1^)		Uncertainty (mm Year^−1^)		Res. Velocity (mm Year^−1^) Eurasia		Res. Velocity (mm Year^−1^) Errachidia (Nubia)	
				East	North	East	North	East	North	East	North
BENI	32.3768	−6.3186	587.1	15.8	17.3	±0.5	±0.6	−4.6	1.2	0.1	−1.1
CEU1	35.8920	−5.3064	52.4	15.2	17.2	±0.4	±0.4	−4.5	1.1	0.2	−1.2
ERRA	31.9388	−4.4561	1104.1	16.1	18.4	±0.6	±0.7	−4.7	2.3	0.0	0.0
RABT	33.9981	−6.8543	90.1	15.8	19.1	±0.5	±0.6	−4.1	3.1	0.6	0.8
TAZA	34.2295	−3.9964	523.5	16.1	19.0	±0.9	±0.9	−4.2	2.9	0.5	0.6

## Data Availability

The data are included in Table 1 of this paper.

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
