# Peer review of "cGPS Record of Active Extension in Moroccan Meseta and Shortening in Atlasic Chains under the Eurasia-Nubia Convergence"

_sensors, 2023, doi:10.3390/s23104846_

Round 1
Reviewer 1 Report
In this paper, the new cGPS data provide reliable support for understanding the geodynamic mechanism that built the prominent Atlasic Cordillera and reveal the heterogeneous present-day behavior of the Eurasia-Nubia collisional setting.
Generally, this paper is very interesting and well-written. This reviewer thinks after a round of revision, it can be accepted.
1. More sentences can be added in Abstract.
2. What is the main contribution and novelty of this paper? It should be highlighted.
3. Introduction can be longer and more sentences can be added.
4. To make the topic of the paper more consistent with the journal, these most related references about “SENSORS” and data processing can be added.
Experimental study on adaptive-passive tuned mass damper with variable stiffness for vertical human-induced vibration control. Engineering Structures, 2023, 280: 115714; The main development of this paper to “SENSORS” should be highlighted.
5. What can be found from Figure 3?
6. What does Figure 5 mean?
7. What is the difference between these figures?
8. Key points should be summarized in Conclusions.
9. According to the fourth question, references can be updated.
Reviewer 2 Report
Dear authors,
You present an interesting paper that can help to understand the geodynamic mechanism of the Atlasic mountain range.
Maybe, the geological setting is too much extense and could be shortened.
In spite of being published, I miss the details and workflows of process, specially the ones dealing with GPS calculations. Please include them.
I recommend to improve as well the maps, since they are difficult to read.
Please find comments below.
Regards.
S16: CSIC-UGR, SPain
Please correct to Spain.
S21: The cGPS 21 network reveals 1 mm yr-1 N-S shortening
It is the first time that the term appears. Please explain what CGPS means. (It does not appear until S70)
S42: in cordilleras compared to classical satellite and surface
Maybe mountain range?
S52: Figure 1. Geological location of the study area. (a) Simplified tectonic location of the study area.
Red texts and symbologies are difficult to read. A Halo or different symbology could improve visualization.
Some names are in Spanish such as Rif Cordillera (mountain range) of central meseta (plateau). Please correct it.
The map has different fonts, sizes and some are bolded. Please homogenize all, and if they are associated to different classes, please include it in the legend.
Interior grids with latitude and longitude are difficult to read. Please move them outside the box.
S54: cordilleras at the northwesternmost mar
Please correct cordilleras, and separe the word most.
S117: Figure 2. Seismicity and main faults of the study area
The map is messy and It is difficult to interpret.
Red texts and symbologies are difficult to read. A Halo or different symbology could improve visualization.
Include the yellow points to the legend.
The black points should be generalized to polygons in some areas where the concentration is high (North area)
Some names are in Spanish such as Rif Cordillera (mountain range) of central meseta (plateau). Please correct it.
The map has different fonts, sizes and some are bolded. Please homogenize all, and if they are associated to different classes, please include it in the legend.
It is difficult to understand the relationship between the colour and B&W map (I know it is the same area)
Please include in the legend the B&W balls (and maybe reduce the size) explaining what are they.
S228: Several approaches to process GPS data were carried out using different software [10]
Despite appearing in the paper, as they are important for the paper, please include them in the paper.
S233: The cGPS data processing followed the standard method used by the University of Jaen [36].
This method should be briefly explained and the workflow included as it is fundamental for the paper.
S234: All the options used at UJA to process cGPS data are shown in [10].
Processing options should be included in the paper as they as fundamental for the research.
S236: Figure 3. Close-up view of the study area that includes the residual CGPS velocities with respect…
The same issues than the previous.
Red texts are difficult to read.
Please increase the resolution of the pictures (in order the artifacts to disappear).
If you want to use red colours, maybe you should change the colours of the polygons to gray-levels or texture.
S241: After processing all the daily data files with Bernese in the ITRF2005 reference frame…
Please cite it, and include processing details (observations, models, adjustment, accuracies, etc) since Bernese is not a “black-box” software as other commercials.
S243: the software NEVE
Please cite it.
S244: the complete stochastic model
Despite being referenced, please show it as it is relevant.
S268: Figure 4.
The same issues than the previous.
Red texts are difficult to read.
Maybe in this and the following pictures, the polygons with the layers are redundant and can be simplified to show the results, hence, the error ellipses and the trace f the figure 6, that now is difficult to read.
S390: Author Contributions
You can use intials instead of full names.
Reviewer 3 Report
The manuscript research cGPS record of active extension in Moroccan Meseta and 2 shortening in Atlasic chains under the Eurasia-Nubia 3 convergence. The Introduction is well structured; the region's geological, seismological, and tectonics review covers the aspect well enough.
However, I want to comment on some critical issues.
- The methodology section is poorly described you said that you used Bernese to process cGPS data and NEVE software. Please describe the methodology that you use. Also, I don’t see any time series. Please add the time series plotting of each station.
- Did you take any seasonality analysis of cGPS time series? Can you demonstrate the method and results? Also, the GPS time series is affected by color noise; how you deal with such an effect on time series? Mainly those two analyses could under or overestimate the velocities of the stations.
- The pre-processing of cGPS is crucial for cGPS quality data. Did you use any pre-processing like multipath or SNR to cGPS rinex files?
With regards
The reviewer
Reviewer 4 Report
My comments:
1. abstact, all acronyms must be explained, e.g. GPS, NNE, SSW
2. introduction, please underline the novelty of paper.
3. main body of text, all acronyms must be explained if their used first time.
4. line 69,please add comma after [10].
5. table 1. please add information about height of stations.
Reviewer 5 Report
REVISION MANUSCRIPT Sensors- 2214888:
“cGPS record of active extension in Moroccan Meseta and shortening in Atlasic chains under the Eurasia-Nubia convergence”
Reviewer #:
General comments:
The authors proposed a study based on the use of 5 continuous GPS stations (installed in the area in 2009) to monitor the northwest-southeast convergence of the Eurasian and Nubian (African) plates in the western Mediterranean region that propagates inside the Nubian plate and affects the Moroccan Meseta and the neighboring Atlasic belt. Therefore, I found the research interesting. However, some suggestions and major revisions need to be applied in order to consider for publication.
Specific comments:
In the introduction, it would be beneficial to include more background information from other studies related to the study area, nearby areas, or similar works. This would provide context for the current study and highlight its novelty.
Regarding the GPS processing part, additional information on the processing, analysis, and error handling should be included. It would be helpful to detail the methodology used for cleaning, offset management, handling outliers, detecting seasonal signals, and any other relevant steps. Furthermore, a description of the time series analysis and how errors were managed should be provided.
Overall, the manuscript would benefit from additional information on the processing and error handling procedures used in the study. Additionally, it would be valuable to incorporate information about GPS background studies in the introduction. The authors should also address the errors indicated in the figures.
Modifications:
In line 21, provide the numerical value for the degree of error.
In line 20-26, change "geodetic" to "geological".
In line 31, "Eurasia-Nubia collisional setting" can be changed to "Eurasia-Nubia collisional boundary" or "Eurasia-Nubia convergent boundary" or "Eurasia-Nubia plate boundary".
In line 44, change "revelation" to "obtain GPS positions with velocity differences" or "deformation rates".
In line 209, change "anchored" to "located".
In line 221, change "against the Middle-Atlasic basement" to "along the boundary with the Middle-Atlasic basement".
In line 234, show all the options or at least the most important.
In line 241, specify how positioning is performed in Bernese. Do you use Precise Point Positioning solution or double-difference solution?
In lines 249, 253, mention the residual and absolute velocities in the same sentence since you refer to the same table.
In figure 3, it is not clear what the "The trace of Figure 3 is also depicted." part refers to.
In figure 3, change "The trace of Figure 6 is also depicted." to "The trace of Figure 6 is represented as a red line on the map.".
In figures 3, 4, the velocity scale appears twice.
In figures 3, 4, 5, Rabat station appears in the legend.
In figures 3, 4, 5, describe the figure in the figure caption (the lines, colors, etc.) from the legend, and in the text, mention the description that you already had because the figure caption is very repetitive.
Round 2
Reviewer 1 Report
accept
Reviewer 2 Report
Dear authors,
You present an interesting paper that can help to understand the geodynamic mechanism of the Atlasic mountain range.
The paper has improved significantly from the previous version and the discussion and conclusions, for me, are fine.
Still, I would improve the part of workflow process (source data) and cartographic representation since some figures are difficult to read and/or interpret. I leave you comments regarding to it.
Congratulations for your efforts.
Regards.

Reviewer 3 Report
Dear author thank you for the updated version of your manuscript.
The reviewer
Reviewer 5 Report
REVISION MANUSCRIPT Sensors- 2214888v2:
“cGPS record of active extension in Moroccan Meseta and shortening in Atlasic chains under the Eurasia-Nubia convergence”
Reviewer #:
General Comments:
Thank you very much for considering me to review this manuscript again! I see that the authors took into account all my previous observations!
The authors worked so hard and well on the manuscript revisions and now I think their paper is suitable for publication!
